# AutoNFS: Automatic Neural Feature Selection

## Abstract

Feature selection (FS) is a fundamental challenge in machine learning, particularly for high-dimensional tabular data, where interpretability and computational efficiency are critical. Existing FS methods often cannot automatically detect the number of attributes required to solve a given task and involve user intervention or model retraining with different feature budgets. Additionally, they either neglect feature relationships (filter methods) or require time-consuming optimization (wrapper and embedded methods). To address these limitations, we propose AutoNFS, which combines the FS module based on Gumbel-Sigmoid sampling with a predictive model evaluating the relevance of the selected attributes. The model is trained end-to-end using a differentiable loss and automatically determines the minimal set of features essential to solve a given downstream task. Unlike existing approaches, AutoNFS achieves a nearly constant computational overhead regardless of input dimensionality, making it scalable to large data spaces. We evaluate AutoNFS on well-established classification and regression benchmarks as well as real-world metagenomic datasets. The results show that AutoNFS consistently outperforms both the classical and neural FS methods while selecting significantly fewer features. We share our implementation of AutoNFS at
`https://anonymous.4open.science/r/AutoNFS-8753`

## 1 Introduction

Feature selection (FS) remains a long-standing challenge in machine learning and data analysis, particularly for high-dimensional tabular datasets, where interpretability and efficiency are crucial Theng & Bhoyar (2024); Dhal & Azad (2022). In practice, such datasets are often constructed by aggregating all available features or by manually engineering additional ones, which frequently leads to an excessive number of variables, many of which contribute little to downstream tasks. FS addresses this issue by identifying and removing redundant or irrelevant features, thereby improving the interpretability of the model, reducing complexity, and providing clearer insights. Furthermore, training a subsequent prediction model on reduced data helps mitigate model overfitting, reduce variance, and often improve predictive performance.

Existing FS approaches can be broadly categorized into filter Yu & Liu (2004); Śmieja et al. (2014), wrapper Kohavi & John (1997); Maldonado & Weber (2009), and embedded methods Tibshirani (1996b); Zou & Hastie (2005), each with inherent limitations. Filter methods rank features according to statistical relevance but remain independent of the learning model, potentially overlooking complex feature interactions. Wrapper methods iteratively select features using the predictive performance of a model as a criterion, but suffer from high computational costs. Embedded methods, such as L1 regularization or attention-based mechanisms, integrate FS within the learning process but may introduce instability or lack fine-grained control over feature importance. The computational cost of most FS algorithms grows rapidly with the number of input dimensions, making them inefficient for large datasets Tan et al. (2014). Additionally, the number of selected features is usually treated as a user-defined hyperparameter; an inappropriate choice can lead to suboptimal performance and require multiple retrainings.

To address these limitations, we propose **AutoNFS**, a neural network for efficient and automatic FS. AutoNFS is a fully differentiable approach, consisting of two networks trained end-to-end (Figure 1). The masking network generates a mask that indicates selected features using temperature-

controlled Gumbel-Sigmoid sampling Maddison et al. (2017); Jang et al. (2017a), while the target network is a predictive model to evaluate their relevance in a downstream task. Unlike existing methods, where the user must specify the desired number of features, AutoNFS automatically determines the minimal subset of features sufficient for the downstream task through a penalty loss component. Moreover, by designing AutoNFS as a modern neural network, it maintains almost constant computational overhead regardless of the dimensionality of the data, making it highly scalable in high dimensions.

We evaluate AutoNFS on well-established classification and regression benchmarks with three scenarios of adding corrupted features Cherepanova et al. (2023). Our experiments demonstrate that AutoNFS consistently outperforms existing techniques while selecting significantly fewer features (Figures 2 and 3). These results are supplemented with the evaluation of AutoNFS in real-world metagenomic datasets (Table 2), analysis of its computational complexity (Figures 4a and 4b) and the visualization of its interpretability in the example of MNIST dataset (Figures 7 and 8).

Our contributions can be summarized as follows.

- We propose AutoNFS, a novel neural network for end-to-end FS, leveraging Gumbel-Sigmoid relaxation and a regularization term that penalizes the number of selected features.
- We show that AutoNFS automatically identifies a minimal yet sufficient subset of features, achieving a nearly constant computational overhead regardless of the input dimensionality, making it scalable for high-dimensional data.
- We validate our approach on well-established OpenML-based benchmarks for FS showing its advantage over related methods. In addition, it is examined on real-world metagenomic datasets, highlighting its effectiveness in high-dimensional biological data analysis.

## 2 RELATED WORK

In Cheng (2024), the importance of FS is reviewed broadly, focusing on filter, wrapper, and embedded methods. Similar surveys have emphasized that the basic taxonomy remains relevant, but must now account for the issues of scalability, fairness, and interpretability in modern high-dimensional data analysis Guyon & Elisseeff (2003); Kohavi & John (1997); Chandrashekar & Sahin (2014); Brown et al. (2012). Due to the page limit, we refer the reader to Appendix A for a detailed description of the classical methods.

The rise of deep learning has inspired neural approaches to FS Ho et al. (2021). Early attempts penalized input weights or used shallow gating networks Li et al. (2016). Later, continuous relaxations allowed discrete masks to be trained via SGD. Louizos et al. (2017) introduced Hard-Concrete gates for $L_0$ regularization; Yamada et al. (2020b) proposed Stochastic Gates (STG); and Balın et al. (2019) designed Concrete Autoencoders that explicitly reconstruct inputs from a subset of features. INVASE Yoon et al. (2018) went further, training an instance-specific selector and predictor in tandem. LassoNet Lemhadri et al. (2021) enforced a hierarchical coupling between a linear skip and deep features to guarantee consistency. Attention mechanisms in Transformers have also been used as feature selectors, but their explanatory validity is contested Serrano & Smith (2019); Jain & Wallace (2019); Gorishniy et al. (2023).

Our work builds on this differentiable line. The technical foundation comes from the Gumbel–Softmax trick Jang et al. (2017a); Maddison et al. (2017), which provides low-variance gradients for sampling. This idea has been extended to subset selection through Gumbel-Top-$k$ Kool et al. (2019), continuous relaxations for sampling without replacement Xie & Ermon (2019), and differentiable sorting operators Blondel et al. (2020). Strypsteen & Bertrand (2024) proposed Conditional Gumbel–Softmax to incorporate structural constraints into FS, such as sensor topologies. Unlike these, AutoNFS addresses unconstrained tabular data and eliminates the need to specify the number of features, letting it emerge from optimization through a cardinality penalty.

Another important line of work studies the acquisition of features *dynamic*, where features have costs and are revealed sequentially. Recent methods query features conditioned on previously observed values Covert et al. (2023); Yasuda et al. (2023), or use reinforcement learning to optimize acquisition policies (e.g., EDDI, budgeted classification) Ma et al. (2019); Janisch et al. (2019); Trapeznikov & Saligrama (2013). These methods are attractive when data acquisition is expensive

(medical tests, sensor readings), but they solve a different problem than ours: we focus on learning a single global mask that amortizes selection across all samples, making inference fast and predictable.

Finally, reliability and fairness in FS have also been addressed. Knockoff-based methods provide false discovery rate control Barber & Candès (2015); Romano et al. (2019), while stability selection explicitly balances sparsity and robustness Meinshausen & Buehlmann (2009). Greedy and OMP-style selectors have been extended to guarantee approximation bounds and fairness in large-scale problems Quinzan et al. (2023). These approaches focus on statistical guarantees, while our method emphasizes efficiency and scalability in neural training.

## 3 THE PROPOSED MODEL

In this section, we introduce AutoNFS, a neural network approach for automatic selection of features, which are relevant for a given machine learning task. First, we give a brief overview of AutoNFS. Next, we describe its main building blocks. Finally, we summarize the training algorithm and the inference phase.

### 3.1 OVERVIEW OF AUTONFS

AutoNFS is a neural network that incorporates features selection into a process of learning a predictive model. It retrieves a variable-size subset of attributes that are the most informative for solving a given classification or regression task.

The architecture of AutoNFS consists of two components: *masking and task networks*, see Figure 1. While the masking network generates a mask representing selected features, the task network solves the underlying task using the indicated attributes. The loss function of AutoNFS combines cross-entropy (for classification) or mean square error (for regression) with the penalty term, which encourages the model to minimize the number of selected features. In consequence, the task network plays the role of a discriminator, which verifies the usefulness of the features chosen for a given task.

In contrast to traditional methods for FS, which iteratively add or reduce attributes, AutoNFS uses a differentiable mechanism to learn a mask based on the Gumbel-Sigmoid relaxation of the discrete distribution Jang et al. (2017a); Maddison et al. (2017). This design ensures that the computational time remains nearly constant regardless of the input dimensionality, making it particularly efficient for high-dimensional data.

### 3.2 MASKING NETWORK

The masking network $f : \mathbb{R}^{D_e} \to \mathbb{R}^D$ is responsible for generating a mask that indicates features selected for a given dataset $\{(x_i, y_i)\}_{i=1}^N \subset \mathbb{R}^D$. Given a randomly initialized input embedding $e \in \mathbb{R}^{D_e}$, the network $f$ outputs $D$-dimensional vector $w = f_\phi(e) \in \mathbb{R}^D$, which determines the mask. More precisely, the output vector $w = (w_1, \ldots, w_D)$ is transformed via a sequence of $D$ Gumbel-Sigmoid functions to the (non-binary) mask vector $m = (m_1, \ldots, m_D)$, where $m_i = GS(w_i; \tau)$ is given by the Gumbel-Sigmoid function with the temperature parameter $\tau > 0$. Let us recall that the Gumbel-Sigmoid function is given by:

$$ \text{GS}(w_i; \tau) = \sigma\left(\frac{w_i + g_i}{\tau}\right), $$

where $g_i \sim -\log(-\log(u))$ with $u \sim \text{Uniform}(0, 1)$ is the Gumbel noise, $\sigma$ is the sigmoid function, and $\tau > 0$ is the temperature parameter.

For $\tau > 0$, the mask $m = (m_1, \ldots, m_D)$ sampled from the Gumbel-Sigmoid distribution can take a continuous (non-binary) form. As $\tau$ decreases, the mask approaches the binary vector, which represents the final discrete mask. Slow decrease of the temperature $\tau$ allows the model to learn the optimal mask during network training.

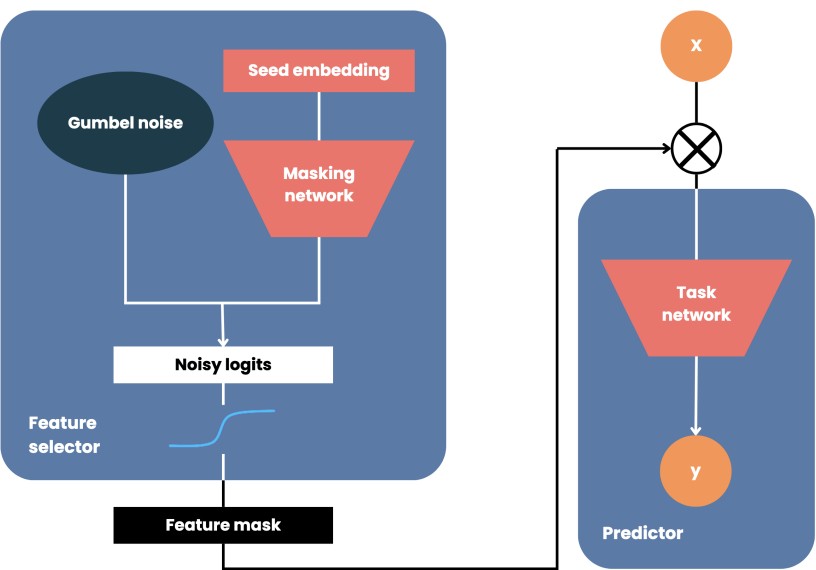

Figure 1: The architecture of AutoNFS consists of two parts: masking and task network. The masking network creates a mask representing selected features, while the target network validates these features on a downstream task. The model is trained end-to-end using a differentiable loss and automatically determines the number of output features.

### 3.3 TASK NETWORK

To learn the optimal mask, we need to verify whether it is informative for the underlying task (e.g. classification). To this end, we first apply a mask $m$ to the input example $x$, by element-wise multiplication $x_m = m \odot x$. Next, we feed a task network $g : \mathbb{R}^D \to Y$ with $x_m$ to obtain the final output $g(x_m)$. The relevance of features selected by $m$ is quantified by the cross-entropy or mean-square loss denoted by $\mathcal{L}_{task}(y; g(x_m))$. Furthermore, to encourage the model to eliminate redundant features, we penalize the model for every added attribute by:

$$\mathcal{L}_{select} = \frac{1}{D} \sum_{j=1}^{D} m_j.$$

The complete loss function is then given by:

$$\mathcal{L}_{total} = \mathcal{L}_{task} + \lambda \mathcal{L}_{select},$$

where $\lambda$ is hyperparameter. We experimentally verified that using a constant value $\lambda = 1$ gives satisfactory results across datasets. Thanks to the Gumbel-Sigmoid relaxation of the discrete mask distribution, we can learn the mask during end-to-end differentiable training.

### 3.4 TRAINING PROCESS

Let us summarize the training algorithm described in Algorithm 1. Training starts with a fixed temperature $\tau = \tau_0$ and a randomly initialized embedding $e$. Given an embedding $e$, the masking network $f$ returns a mask vector $m = (m_1, \ldots, m_D)$ using the Gumbel-Sigmoid functions. Each continuous mask vector $m$ sampled from Gumbel-Sigmoid is then applied to a mini-batch $\mathcal{B}$ to construct the reduced vectors $x_m = m \odot x$, for $x \in \mathcal{B}$. This vector goes to the task network $g$, which returns the response for a given task $g(x_m)$. The loss function $\mathcal{L}_{total}$ is calculated and the gradient is propagated to: (1) embedding vector $e$, (2) weights of $f$ and $g$. In particular, by learning the embedding vector $e$ and the parameters of $f$, we optimize the mask vector.

A critical aspect of our algorithm is the temperature annealing schedule. We begin with a high temperature ($\tau = 2.0$), which produces soft masks that allow gradient flow to all features. As

---

**Algorithm 1** AutoNFS training procedure for classification

---

1: **Input:** Dataset $\mathcal{D} = \{(\mathbf{x}_i, \mathbf{y}_i)\}_{i=1}^N$, batch size $B$, initial temperature $\tau_0 = 2.0$, decay rate $\alpha = 0.997$, total epochs $E$, FS balance parameter $\lambda$
2: **Initialize:** Embedding vector $\mathbf{e} \in \mathbb{R}^{d_e}$, masking network $f_\phi$, task network $g_\theta$
3: $\tau \leftarrow \tau_0$
4: **for** epoch $= 1$ to $E$ **do**
5:     **for** each mini-batch $\mathcal{B} = \{(\mathbf{x}_i, \mathbf{y}_i)\}_{i=1}^B \subset \mathcal{D}$ **do**
6:        $\mathbf{w} \leftarrow f_\phi(\mathbf{e})$            ▷ Compute logits for feature mask
7:        $\mathbf{g} \leftarrow -\log(-\log(\mathbf{u}))$, where $\mathbf{u} \sim \text{Unif}(0, 1)$            ▷ Sample Gumbel noise
8:        $\mathbf{m} \leftarrow \sigma\left((\mathbf{w} + \mathbf{g})/\tau\right)$            ▷ Generate mask via Gumbel-Sigmoid
9:        $\mathbf{X} \leftarrow \{\mathbf{x}_i\}_{i=1}^B$
10:        $\mathbf{X}_{\text{masked}} \leftarrow \mathbf{X} \odot \mathbf{m}$            ▷ Mask input features
11:        $\hat{\mathbf{Y}} \leftarrow g_\theta(\mathbf{X}_{\text{masked}})$            ▷ Forward pass through task network
12:
13:        $\mathcal{L}_{\text{task}} \leftarrow -\sum_{i=1}^B \sum_{c=1}^C y_{i,c} \log(\hat{y}_{i,c})$
14:        $\mathcal{L}_{\text{select}} \leftarrow \frac{1}{D} \sum_{j=1}^D m_j$
15:        $\mathcal{L}_{\text{total}} \leftarrow \mathcal{L}_{\text{task}} + \lambda \cdot \mathcal{L}_{\text{select}}$
16:
17:        $\mathbf{e} \leftarrow \mathbf{e} - \eta_1 \nabla_\mathbf{e} \mathcal{L}_{\text{total}}$            ▷ Update embedding
18:        $\phi \leftarrow \phi - \eta_1 \nabla_\phi \mathcal{L}_{\text{total}}$            ▷ Update masking network
19:        $\theta \leftarrow \theta - \eta_2 \nabla_\theta \mathcal{L}_{\text{total}}$            ▷ Update task network
20:     **end for**
21:     $\tau \leftarrow \tau \cdot \alpha$            ▷ Anneal temperature
22: **end for**

---

training progresses, the temperature decays exponentially (typically with $\alpha = 0.997$), causing the masks to become increasingly binary. This gradual transition serves multiple purposes:

- It allows the network to initially explore the full feature space.
- It enables progressive commitment to more discrete FS decisions.
- It leads to convergence on a nearly binary FS mask at the end of training.

The annealing process effectively functions as a curriculum, starting with easier optimization (continuous selection) and progressively transitioning to harder optimization (discrete selection). This process is related to exploration-exploitation trade-off, which parallels fundamental concepts in reinforcement learning (see Appendix B detailed discussion).

### 3.5 FEATURE IMPORTANCE QUANTIFICATION

After training, we quantify the importance of each feature by directly applying the learned selection mechanism with hard Gumbel-Sigmoid activation:

1. Calculating the feature logits of the trained embedding: $\mathbf{w} = \mathbf{f}_\phi(\mathbf{e})$.
2. Applying a hard threshold, that is, if $\sigma(w_i) > 0.5$, then $m_i = 1$, else $m_i = 0$.
3. Interpreting the resulting binary vector $\mathbf{m} = (m_1, \ldots, m_D)$ as the mask for feature selection.

This process produces a deterministic FS that clearly identifies relevant features for the task. Since our FS mechanism is parameterized by a single embedding vector that is independent of specific input examples, the selected features remain constant throughout the dataset. The resulting binary mask can be directly used to filter features, or features can be ranked by their logit values when a specific top-$k$ selection is desired. Importantly, since the selection mechanism was jointly optimized with the task objective, the selected features capture both individual importance and interactive effects relevant to the specific task.

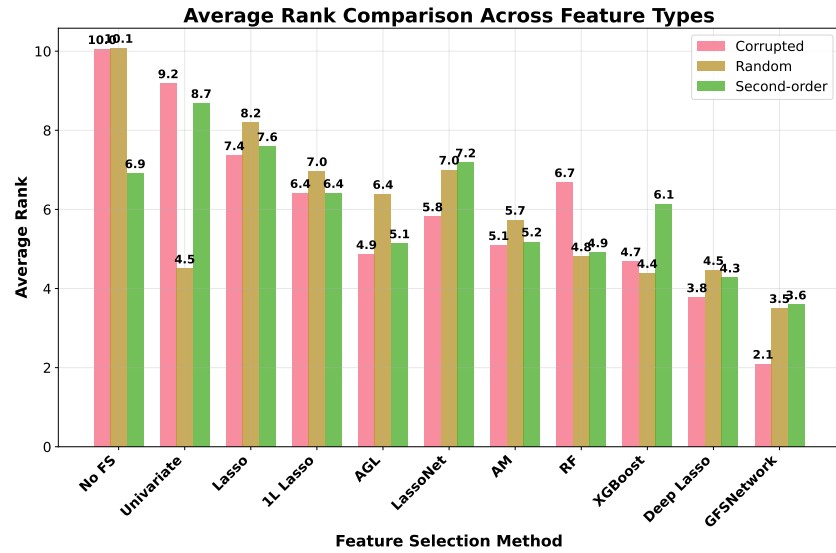

Figure 2: Average ranking of FS methods for three types of features corruption shows that AutoNFS consistently outperforms all competitive methods.

## 4 EXPERIMENTS

To evaluate the effectiveness of AutoNFS, we conducted extensive experiments across multiple datasets (standard OpenML data and high-dimensional metagenomic datasets) and compared our approach with state-of-the-art FS methods. We verify the performance of the model and inspect the importance of selected attributes. Furthermore, we analyze the computational efficiency of our method compared to existing approaches and the influence of the parameter $\lambda$ on the behavior of the algorithm Appendix F. We also provide further insight into the interpretability of the selected features in the example of MNIST, which can be found in Appendix G.

### 4.1 BENCHMARK DATASETS

Table 1: Summary of datasets (left) and the number of attributes selected by AutoNFS under three considered scenarios (right). It is evident that AutoNFS not only eliminate auxiliary noisy features but also drastically reduces the number of the original attributes.

| Dataset | Dataset Statistics | | | Features Selected by AutoNFS | | |
| | Samples | Classes | Features | Random Features | Corrupted Features | Second-order Features |
|---|---|---|---|---|---|---|
| AL (aloi) | 108 000 | 1000 | 128 | 65 | 65 | 69 |
| CH (california) | 20 640 | regression | 8 | 5 | 5 | 3 |
| EY (eye) | 10 936 | 3 | 26 | 8 | 11 | 12 |
| GE (gesture) | 9 873 | 5 | 32 | 11 | 16 | 22 |
| HE (helena) | 65 196 | 100 | 27 | 15 | 14 | 16 |
| HI (higgs_small) | 98 050 | 2 | 28 | 14 | 14 | 14 |
| HO (house) | 22 784 | regression | 16 | 10 | 10 | 9 |
| JA (jannis) | 83 733 | 4 | 54 | 17 | 16 | 18 |
| MI (microsoft) | 1 200 192 | regression | 136 | 47 | 61 | 42 |
| OT (otto) | 61 878 | 9 | 93 | 78 | 67 | 76 |
| YE (year) | 515 345 | regression | 90 | 69 | 28 | 29 |

**Experimental setup** We follow a recent benchmark introduced in Cherepanova et al. (2023). The reported results were achieved by extending their code base with AutoNFS.

The benchmark consists of three scenarios applied to 11 datasets (see LHS of Table 1). In each, a given dataset is corrupted by adding auxiliary features: (1) fully random features, (2) original features corrupted with Gaussian noise, and (3) a set of second-order features created by multiplying randomly selected features from the original dataset. We analyze a scenario in which 50% of the features in each dataset were artificially created. By applying FS algorithms, we aim to eliminate redundant features without compromising the predictive power of the representation.

We compared AutoNFS with 10 established FS methods. All methods use MLP as a downstream classifier. We refer the reader to Appendix C for further details of the experimental setup.

For each dataset and method, we compute performance metrics specific to the task (accuracy for classification, negative mean squared error for regression). We also report the mean rank across datasets to provide an overall performance assessment.

**Predictive performance**    The ranking summary of the results presented in Figure 2 shows an impressive performance of AutoNFS in each scenario. While the highest advantage of AutoNFS is observed for the case of features corrupted by Gaussian noise (average rank 2.1), in the remaining two scenarios (random and second-order features) AutoNFS still achieves the best ranks, beating the next competitors by 0.9 and 0.7 ranking points, respectively. It is important to note that all baseline methods select the same number of features as were in the initial representation (before corruption), whereas our method automatically chooses a much smaller subset of the most relevant features, see the RHS of Table 1. As a result, AutoNFS consistently achieves competitive or superior performance while using significantly fewer features, highlighting its practical advantage. Detailed results presented in Tables 3 to 5 show that our algorithm obtains the highest or joint-highest scores on most datasets, demonstrating consistent and strong performance.

**Analysis of selected features**    In addition to predictive performance on downstream tasks, we analyze how the selected attributes match the original features (before adding auxiliary features). Figure 3a shows that AutoNFS achieves zero misselection errors for random and corrupted features and maintains low error rates of 0.17 for second-order features. It is important to note that the selection of features outside the original attributes in the latter case is acceptable since additional features were created by multiplying the original features. In consequence, these extra features may sometimes carry even more information than the individual original attributes. The application of the representation created by the baseline methods resulted in significantly higher misselection errors.

Figure 3b presents the average predictive power of the individual features. More precisely, we measure how much predictive performance decreases when we remove one of the selected features. As can be seen, the average decrease for AutoNFS is equal to 0.313, which means that the returned set cannot be further reduced without affecting predictive performance. This demonstrates the superior precision of AutoNFS in identifying relevant features while automatically determining the optimal number to select.

In general, these findings confirm that AutoNFS is broadly applicable to a wide range of machine learning tasks, including both classification and regression, while offering strong and reliable performance in various feature noise scenarios.

## 4.2 METAGENOMIC DATASET ANALYSIS

To evaluate AutoNFS's effectiveness in real-world high-dimensional biological data, we applied it to 24 metagenomic datasets obtained from Curated Metagenomics Data Pasolli et al. (2017). These datasets represent a particularly challenging domain with high feature dimensionality (308-718 features) and complex biological interactions. In this experiment, we additionally verify how the constructed representation is useful for two types of downstream classifiers: MLP and Random Forest (RF).

The results presented in Table 2 demonstrate that, on average, AutoNFS maintains predictive performance on downstream tasks while drastically reducing feature dimensionality (AutoNFS selected only 7.7% of the original features). In the case of MLP, AutoNFS achieved 0.7 improvements in pp accuracy, while for RF the improvement increased to 1.2 pp. This means that the high predictive performance of the representation generated by AutoNFS is independent of a downstream classifier.

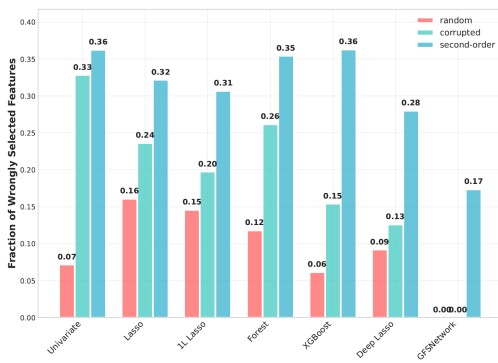

(a) Feature misselection errors. In 2 out of 3 corruption scenarios, AutoNFS selects only features from the original ones, presenting the best performance in all cases.

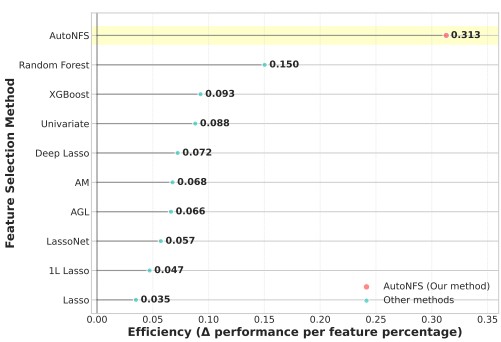

(b) Average predictive power of the selected variables in the case of random feature corruption shows that AutoNFS selects the most essential features – the average performance of downstream model would decrease by 0.313 if any of features selected by AutoNFS were eliminated.

Figure 3: Analysis of features selected by examined methods.

Table 2: Performance on metagenomic data reduced with AutoNFS. Although AutoNFS heavily reduces data dimensionality, it does not lead to the deterioration of the results on average. Each dataset's name is derived from the first author's surname and the year of publication.

| dataset | MLP on full data | MLP on AutoNFS | RF on full data | RF on AutoNFS | Original dim. | Reduced dim. |
|---|---|---|---|---|---|---|
| NielsenHB_2014 | 0.613 | **0.643** | **0.711** | 0.634 | 370 | 33 |
| WirbelJ_2018 | 0.558 | **0.571** | 0.776 | **0.821** | 639 | 32 |
| KeohaneDM_2020 | **0.469** | 0.344 | 0.469 | **0.531** | 540 | 37 |
| JieZ_2017 | **0.693** | 0.612 | 0.762 | **0.770** | 308 | 61 |
| FengQ_2015 | **0.662** | 0.607 | 0.833 | **0.889** | 575 | 25 |
| ThomasAM_2019c | 0.582 | **0.664** | 0.627 | **0.764** | 438 | 32 |
| LiJ_2017 | 0.341 | **0.511** | **0.561** | 0.432 | 651 | 43 |
| ZellerG_2014 | 0.614 | 0.614 | 0.652 | **0.871** | 645 | 23 |
| LifeLinesDeep_2016 | 0.513 | **0.546** | **0.500** | **0.500** | 526 | 79 |
| ThomasAM_2018b | **0.686** | 0.614 | **0.586** | **0.586** | 621 | 31 |
| HanniganGD_2017 | 0.467 | **0.633** | **0.817** | 0.533 | 477 | 22 |
| YachidaS_2019 | 0.471 | **0.570** | **0.636** | 0.608 | 480 | 88 |
| ZhuF_2020 | **0.657** | 0.559 | **0.768** | 0.739 | 718 | 33 |
| ThomasAM_2018a | **0.733** | 0.567 | 0.817 | **0.917** | 292 | 24 |
| LiJ_2014 | 0.454 | **0.490** | 0.500 | **0.508** | 503 | 46 |
| LeChatelierE_2013 | **0.551** | 0.521 | 0.549 | **0.620** | 646 | 51 |
| QinN_2014 | 0.746 | **0.815** | 0.833 | **0.855** | 652 | 38 |
| QinJ_2012 | 0.551 | **0.561** | 0.616 | **0.622** | 436 | 59 |
| NagySzakalD_2017 | 0.521 | **0.583** | 0.917 | **0.958** | 519 | 21 |
| YuJ_2015 | **0.653** | 0.417 | **0.674** | 0.646 | 606 | 34 |
| GuptaA_2019 | 0.812 | **0.938** | 0.875 | **0.938** | 683 | 19 |
| VogtmannE_2016 | 0.667 | **0.681** | **0.694** | **0.694** | 381 | 38 |
| AsnicarF_2021 | 0.503 | **0.528** | **0.500** | **0.500** | 537 | 90 |
| RubelMA_2020 | 0.607 | **0.717** | 0.775 | **0.796** | 606 | 26 |
| average | 0.588 | 0.596 | 0.685 | 0.697 | 535 | 41 |

Figure 5 illustrates the process of FS. Observe that AutoNFS deeply explores the space of all features in the training phase and selects the final set of features at the end of the training.

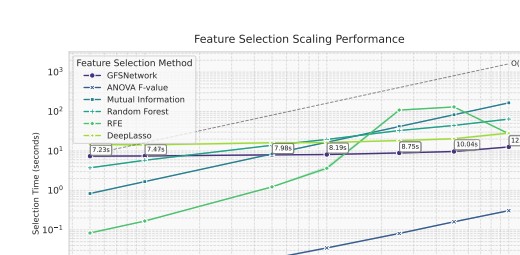 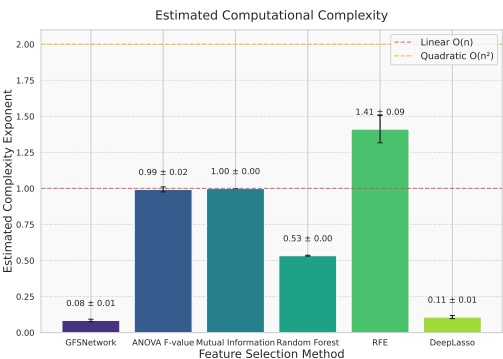

(a) The time requirements of AutoNFS remains almost constant with increasing number of features.

(b) Comparison of estimated complexity exponent for different FS methods as dimensionality changes.

Figure 4: Estimation of time complexity.

### 4.3 COMPUTATIONAL COMPLEXITY ESTIMATION

The estimated computational complexity reveal striking differences between FS methods, see Figure 4a. Denoting time complexity as an exponential function of the number of features $t \approx D^\alpha$, our empirical analysis shows that AutoNFS demonstrates near-constant time scaling ($\alpha \approx 0.08$). Conventional FS methods, such as the ANOVA F value and Mutual Informatio, exhibit linear scaling ($\alpha \approx 1.0$), while Random Forest FS shows sublinear behavior ($\alpha \approx 0.53$). In contrast, Recursive Feature Elimination with Linear SVC demonstrates superlinear scaling ($\alpha \approx 1.41$), causing its performance to degrade more rapidly with increasing feature dimensions.

The confidence intervals over 5 runs (Figure 4b) indicate that these estimates are statistically robust across the dimensionalities tested. This assessment provides compelling evidence for the exceptional efficiency advantage of AutoNFS in high-dimensional FS tasks, with its nearly constant-time behavior representing a significant algorithmic advancement over conventional methods.

## 5 CONCLUSION

We presented AutoNFS, a novel neural architecture for FS in a differentiable end-to-end manner using temperature-controlled Gumbel-Sigmoid sampling. The key innovation lies in its ability to automatically determine not only which features are relevant but also how many features should be retained, a common pain point in traditional FS methods. Whereas most existing techniques require the number of selected features to be manually specified or found through expensive hyperparameter tuning, AutoNFS learns this quantity during training.

Experimental results in synthetic benchmarks and real-world datasets demonstrate that AutoNFS consistently selects fewer features than baselines, without compromising predictive performance. This reduction is beneficial in terms of computational efficiency and interpretability, but also validates the model's ability to avoid overfitting by ignoring redundant or noisy inputs.

Looking ahead, this automatic feature count discovery opens doors for broader applications, such as real-time model compression, adaptive inference, or integration with AutoML frameworks. Moreover, the balance between sparsity and accuracy, controlled through a single $\lambda$ parameter, makes AutoNFS a drop-in replacement for feature selectors in a wide range of tasks.

**Ethics statement.** This paper presents work whose goal is to advance the field of Machine Learning. There are many potential societal consequences of our work, none of which we feel must be highlighted here.

**Reproducibility statement.** We have described all the details and hyperparameters of the proposed approach. We include our codebase as an anonymous repository and will publish it along with the paper.

**LLM statement.** The authors used LLM tools to polish the writing.

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

## A    EXTENDED RELATED WORK

Filter methods typically rely on statistical criteria such as correlation, mutual information, or significance tests. Classical examples include mRMR Peng et al. (2005), Relief and its variants Kononenko (1994); Robnik-Sikonja & Kononenko (2003), or kernel-based criteria like HSIC Lasso Yamada et al. (2014). More recent efforts include measures based on the maximal information coefficient (MIC) to capture non-linear associations Reshef et al. (2011). These methods are computationally efficient and easy to interpret, but they ignore feature interactions and are detached from the final predictive model, which often leads to suboptimal subsets.

Wrapper methods overcome this by iteratively selecting subsets guided by model performance. Classical strategies include sequential forward/backward selection and floating search Pudil et al. (1994), SVM-RFE for ranking genes Guyon et al. (2002), and more recent ensemble-based approaches such as Boruta, which compares importance with permuted shadow features Kursa & Rudnicki (2010). Wrappers usually achieve higher accuracy, but their repeated training makes them infeasible for high-dimensional data or large-scale tasks.

Embedded methods integrate FS directly into the model learning phase. The best known are sparsity-inducing penalties like Lasso Tibshirani (1996a), Elastic Net Zou & Hastie (2005), and Group Lasso Yuan & Lin (2006); Simon et al. (2013). Tree-based ensembles provide another embedded route: feature importance can be derived from Random Forests Breiman (2001) or boosting models like XGBoost and CatBoost Chen & Guestrin (2016); Prokhorenkova et al. (2019). Embedded methods combine efficiency and accuracy, but they are biased toward the structure of the underlying model (linear, tree-based), and may struggle in domains with correlated features. Stability selection was proposed to mitigate these limitations Meinshausen & Buehlmann (2009).

## B    EXPLORATION-EXPLOITATION TRADE-OFF OF AUTONFS

The temperature-controlled sampling enables our model to transition smoothly from exploration (high temperature) to exploitation (low temperature) Jang et al. (2017b); Haarnoja et al. (2018). This exploration-exploitation trade-off parallels fundamental concepts in reinforcement learning (RL) Sutton & Barto (2018). The FS problem can be framed as a contextual multi-armed bandit, where each feature represents an "arm" with an unknown reward distribution Lattimore & Szepesvári (2020); Li et al. (2010). Initially, with high temperature, our model explores the feature space broadly – similar to how RL agents employ $\epsilon$-greedy or softmax policies with high entropy to explore their environment Sutton & Barto (2018); Cesa-Bianchi et al. (2017). As training progresses and the temperature decreases, the model increasingly exploits high-value features, analogous to how RL agents converge toward optimal policies.

The temperature annealing schedule therefore functions as an adaptive exploration strategy, initially permitting broad sampling of the feature space before gradually committing to the most informative features Jang et al. (2017b); Maddison et al. (2017); Kirkpatrick et al. (1983). This approach prevents premature convergence to suboptimal feature subsets, a common challenge in both FS and reinforcement learning Kirkpatrick et al. (1983). Furthermore, end-to-end training with the primary task provides an implicit reward signal that guides the FS policy, where improvements in task performance reinforce the selection of beneficial features through gradient updates to the selection parameters Balın et al. (2019); Yamada et al. (2020a).

## C    EXPERIMENTAL SETUP

**Baseline Methods**    Following Cherepanova et al. (2023), we compared AutoNFS with ten established FS methods:

- No Feature Selection (No FS),
- Univariate Selection: Statistical tests for feature ranking (F-statistics),
- Lasso: L1-regularized linear models,
- 1L Lasso: Single-layer neural network with L1 regularization,
- AGL: Adaptive Group Lasso Ho et al. (2021),
- LassoNet: Neural network with hierarchical sparsity Lemhadri et al. (2021),

- AM: Attention Maps for feature importance Gorishniy et al. (2023),
- RF: Random Forest importance,
- XGBoost: Gradient boosting importance Chen & Guestrin (2016),
- Deep Lasso: Deep neural network with L1 regularization Cherepanova et al. (2023).

**Model Architecture and Hyperparameters**   AutoNFS consists of a 32-dimensional learnable embedding that projects to feature-specific selection logits through a linear layer ($32 \rightarrow$ input_size), followed by a 3-layer task network with architecture input_size $\rightarrow 32 \rightarrow 32 \rightarrow$ output_size using ReLU activations. Hyperparameters were optimized using Optuna Akiba et al. (2019) across epochs $\in \{10, 20, 50, 100, 200, 300, 400\}$, temperature decay $\in \{0.995, 0.997, 0.999\}$, and batch sizes $\in \{32, 64, 128\}$, with target features mode selected from {"raw", "target"}.

We use Adam Kingma & Ba (2017) optimizer with separate learning rates: 4e-3 for the FS component and 3e-4 for the task network. The Gumbel-Sigmoid temperature starts at 2.0 and decays per epoch, while the FS balance parameter $\lambda$ is set to 1.0. This design ensures nearly constant computational overhead regardless of input dimensionality while maintaining effective FS capabilities.

# D   DETAILED FS RESULTS

Tables 3–5 report detailed results of our experimental evaluation on three benchmark scenarios for FS: **random features**, **corrupted features**, and **second-order features**. For each setting, we present classification (accuracy) and regression (negative MSE) performance for all compared methods across multiple datasets. The last column shows the mean rank for each method (lower is better).

All baseline methods (e.g., Univariate, Lasso, RF, XGBoost, Deep Lasso, AM, LassoNet) select the full set of features, while our method (AutoNFS) automatically selects a much smaller, data-driven subset. The best results for each dataset are highlighted in bold.

These tables illustrate that AutoNFS consistently achieves strong or state-of-the-art performance across a wide variety of noise types and dataset structures, while requiring far fewer features than conventional methods. This confirms the robustness and practical effectiveness of our approach under diverse experimental conditions.

Table 3: Classification (accuracy) and regression (negative MSE) performance in the case of random features. Higher values denote better scores.

| FS method | AL | CH | EY | GE | HE | HI | HO | JA | MI | OT | YE | rank |
|---|---|---|---|---|---|---|---|---|---|---|---|---|
| No FS | 0.941 | -0.48 | 0.538 | 0.466 | 0.366 | 0.798 | -0.622 | 0.703 | -0.911 | 0.773 | -0.801 | 10.1 |
| Univariate | **0.96** | -0.447 | 0.575 | 0.515 | 0.379 | 0.811 | **-0.549** | 0.715 | **-0.891** | 0.808 | -0.776 | 4.5 |
| Lasso | 0.949 | -0.454 | 0.547 | 0.458 | 0.38 | 0.812 | -0.599 | 0.715 | -0.907 | 0.805 | -0.787 | 8.2 |
| 1L Lasso | 0.952 | -0.451 | 0.564 | 0.474 | 0.375 | 0.811 | -0.568 | 0.715 | -0.897 | 0.796 | **-0.773** | 7.0 |
| AGL | 0.958 | -0.512 | 0.578 | 0.473 | **0.386** | 0.81 | -0.557 | 0.718 | -0.898 | 0.799 | -0.778 | 6.4 |
| LassoNet | 0.954 | -0.445 | 0.552 | 0.495 | 0.385 | 0.811 | -0.557 | 0.715 | -0.907 | 0.783 | -0.787 | 7.0 |
| AM | 0.953 | -0.444 | 0.554 | 0.498 | 0.382 | 0.813 | -0.566 | 0.722 | -0.904 | 0.801 | -0.777 | 5.7 |
| RF | 0.955 | -0.453 | 0.589 | **0.594** | **0.386** | 0.814 | -0.572 | 0.72 | -0.904 | 0.806 | -0.786 | 4.8 |
| XGBoost | 0.956 | -0.444 | 0.59 | 0.502 | 0.385 | 0.812 | -0.56 | 0.72 | -0.893 | 0.805 | -0.777 | 4.4 |
| Deep Lasso | 0.959 | -0.443 | 0.573 | 0.485 | 0.383 | 0.814 | **-0.549** | 0.72 | -0.894 | 0.802 | -0.776 | 4.5 |
| AutoNFS | **0.96** | **-0.441** | **0.634** | 0.55 | 0.375 | **0.818** | -0.565 | **0.738** | -0.893 | **0.811** | -0.782 | **3.5** |

# E   FEATURE SPACE EVOLUTION

Figure 5 illustrates the dynamic FS process of AutoNFS through temperature-controlled Gumbel-Sigmoid sampling conducted on metagenomic data (see Section 4.2). The visualization demonstrates how the model transitions from broad feature exploration (high temperature T=5.0) to decisive feature commitment (low temperature T=0.5) during training.

Initially, FS probabilities exhibit high variance and exploration across the entire feature space, with the Gumbel noise enabling stochastic sampling. As temperature anneals exponentially, the selection probabilities converge toward binary decisions. Features ultimately selected by the model (shown in

Table 4: Classification (accuracy) and regression (negative MSE) performance in the case of corrupted features. Higher values denote better scores.

| FS method | AL | CH | EY | GE | HE | HI | HO | JA | MI | OT | YE | rank |
|---|---|---|---|---|---|---|---|---|---|---|---|---|
| No FS | 0.946 | -0.475 | 0.557 | 0.525 | 0.37 | 0.802 | -0.607 | 0.703 | -0.909 | 0.778 | -0.797 | 10.0 |
| Univariate | 0.955 | -0.451 | 0.556 | 0.514 | 0.346 | 0.81 | -0.62 | 0.717 | -0.92 | 0.795 | -0.828 | 9.2 |
| Lasso | 0.955 | -0.449 | 0.548 | 0.512 | 0.382 | 0.813 | -0.602 | 0.713 | -0.903 | 0.796 | -0.795 | 7.4 |
| 1L Lasso | 0.955 | -0.447 | 0.566 | 0.515 | 0.382 | 0.812 | -0.581 | 0.718 | -0.902 | 0.795 | -0.78 | 6.4 |
| AGL | 0.953 | -0.45 | 0.588 | 0.538 | 0.386 | 0.813 | -0.561 | 0.722 | -0.902 | 0.796 | -0.78 | 4.9 |
| LassoNet | 0.955 | -0.452 | 0.57 | 0.556 | 0.382 | 0.811 | -0.551 | 0.719 | -0.905 | 0.795 | -0.777 | 5.8 |
| AM | 0.955 | -0.449 | 0.583 | 0.527 | 0.381 | 0.814 | -0.555 | 0.722 | -0.905 | 0.797 | -0.78 | 5.1 |
| RF | 0.951 | -0.453 | 0.574 | 0.568 | 0.383 | 0.81 | -0.565 | 0.724 | -0.904 | 0.788 | -0.786 | 6.7 |
| XGBoost | 0.954 | -0.454 | 0.583 | 0.51 | 0.385 | 0.815 | -0.553 | 0.722 | **-0.892** | 0.803 | -0.779 | 4.7 |
| Deep Lasso | 0.955 | -0.447 | 0.577 | 0.525 | **0.388** | 0.815 | -0.567 | 0.721 | -0.895 | 0.801 | **-0.776** | 3.8 |
| AutoNFS | **0.957** | **-0.437** | **0.625** | **0.57** | 0.373 | **0.819** | -0.549 | **0.735** | -0.895 | **0.804** | -0.779 | **2.1** |

Table 5: Classification (accuracy) and regression (negative MSE) performance in the case of second-order features. Higher values denote better scores.

| FS method | AL | CH | EY | GE | HE | HI | HO | JA | MI | OT | YE | rank |
|---|---|---|---|---|---|---|---|---|---|---|---|---|
| No FS | 0.96 | -0.443 | 0.631 | 0.605 | 0.383 | 0.811 | -0.549 | 0.719 | -0.891 | 0.8 | -0.786 | 6.9 |
| Univariate | **0.961** | -0.439 | 0.584 | 0.582 | 0.357 | 0.817 | -0.614 | 0.724 | -0.902 | 0.798 | -0.81 | 8.7 |
| Lasso | 0.955 | -0.443 | 0.608 | 0.59 | 0.366 | 0.816 | -0.564 | 0.724 | -0.891 | 0.806 | -0.783 | 7.6 |
| 1L Lasso | 0.959 | -0.445 | 0.634 | 0.571 | 0.38 | 0.815 | -0.565 | 0.728 | **-0.89** | **0.808** | -0.78 | 6.4 |
| AGL | 0.961 | -0.443 | 0.637 | 0.594 | 0.383 | 0.807 | -0.565 | 0.73 | **-0.89** | 0.806 | -0.776 | 5.1 |
| LassoNet | 0.959 | -0.442 | 0.641 | 0.611 | 0.379 | 0.816 | -0.595 | 0.724 | -0.893 | 0.797 | -0.784 | 7.2 |
| AM | **0.961** | -0.439 | 0.622 | 0.604 | 0.381 | **0.819** | -0.566 | 0.73 | -0.892 | 0.802 | -0.778 | 5.2 |
| RF | 0.958 | -0.437 | 0.639 | **0.619** | 0.37 | 0.818 | -0.586 | 0.735 | **-0.89** | 0.801 | -0.781 | 4.9 |
| XGBoost | 0.87 | -0.438 | 0.635 | 0.604 | 0.373 | 0.818 | -0.579 | 0.734 | -0.891 | 0.805 | -0.786 | 6.1 |
| Deep Lasso | **0.961** | -0.441 | **0.648** | 0.6 | **0.384** | 0.815 | -0.572 | 0.733 | **-0.89** | 0.805 | -0.776 | 4.3 |
| AutoNFS | 0.96 | **-0.436** | 0.638 | 0.6 | 0.378 | 0.817 | -0.548 | 0.738 | -0.891 | 0.808 | **-0.775** | **3.6** |

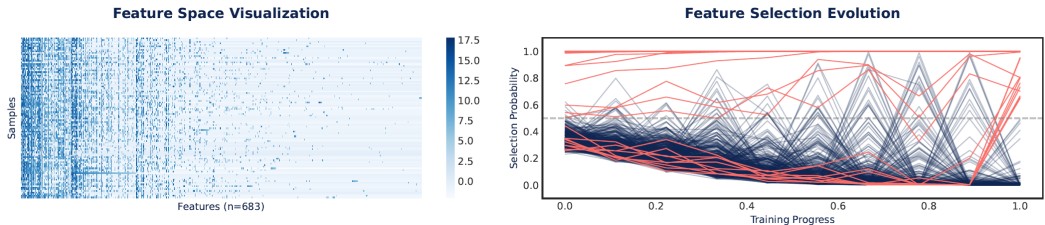

Figure 5: FS analysis showing the feature space representation (left) and selection probability evolution (right). The left heatmap displays the distribution of features across samples, with color intensity indicating feature values. The right evolution plot tracks selection probabilities throughout training progress, highlighting distinct patterns between selected features (red) that maintain high probabilities either from initialization or emerge at later stages, versus non-selected features (blue) that exhibit diminishing selection probabilities over time.

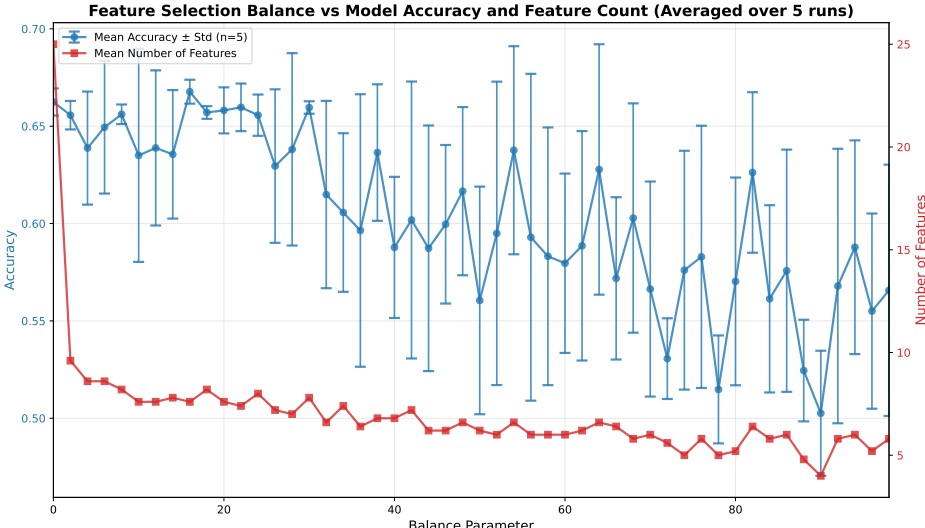

Figure 6: Effect of the balance parameter $\lambda$ on predictive accuracy (blue, left axis) and the number of selected features (red, right axis). When $\lambda = 0$, AutoNFS prioritizes task performance and selects a large number of features. As $\lambda$ increases, the sparsity penalty reduces the number of features while preserving accuracy up to a point. Very high values of $\lambda$ cause over-sparsification, where too few features are selected, leading to performance degradation. Results are averaged over 5 runs with standard deviations shown as error bars.

red) either maintain consistently high probabilities throughout training or emerge during later stages, while non-selected features (shown in blue/dark) exhibit diminishing selection probabilities over time. This evolution process effectively implements a learnable exploration-exploitation strategy, where the model initially considers all features before gradually committing to the most informative subset for the given task.

## F  INFLUENCE OF THE BALANCE PARAMETER

The balance parameter $\lambda$ in AutoNFS controls the trade-off between task performance and feature sparsity through the regularization term $\lambda\mathcal{L}_{select}$ in the total loss function, see Figure 6. When $\lambda = 0$, the model prioritizes the performance of tasks without penalizing the use of features, typically selecting a larger number of features. As $\lambda$ increases, the sparsity penalty becomes more influential, forcing the model to select fewer features while trying to maintain predictive accuracy. However, excessively high values of $\lambda$ lead to over-sparsification, where the model selects too few features to adequately capture the underlying patterns, resulting in performance degradation. This analysis demonstrates the importance of proper tuning of $\lambda$ and highlights how AutoNFS can automatically navigate the accuracy-sparsity trade-off to identify optimal feature subsets in different datasets.

## G  FEATURE SELECTION VISUALIZATION ON MNIST

To provide intuitive insights into how AutoNFS selects discriminative features, we conducted visualization experiments on the MNIST handwritten digit dataset. Figure 7 (left) compares the entropy of the selected vs. non-selected features. It is evident that AutoNFS focuses more on discriminative features (with higher entropy). The mean entropy of selected features equals 1.98 while the mean entropy of all remaining features equals 1.43. Figure 7 (right) illustrates the pixels selected for the MNIST dataset. Clearly, the model pays more attention to the center of the image, ignoring the background regions.

Finally, Figure 8 examines individual selected features (blue) for a sample digit, comparing their class-conditional activation distributions with non-selected pixels (red). Selected features consis-

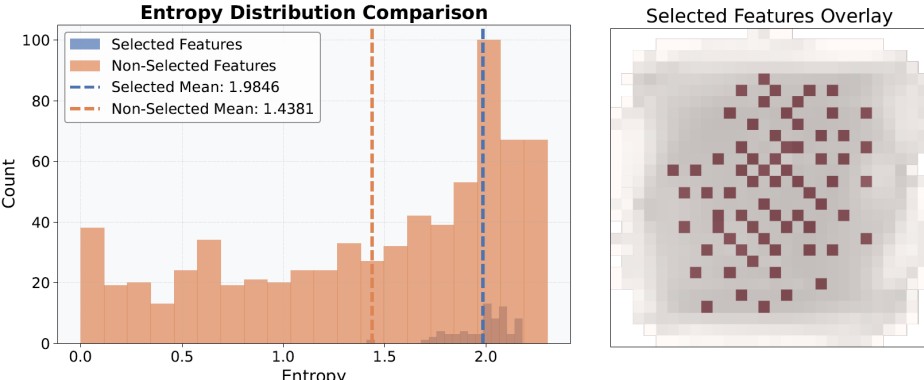

Figure 7: Average entropy of selected features is significantly higher than the entropy of all features, which means that AutoNFS selected features with discriminative potential (left). Moreover, selected pixel are localized in the center region of the image (right).

tently show more discriminative patterns across digit classes, with higher entropy values indicating higher information content. These visualizations demonstrate that AutoNFS selects features in a manner that aligns with human intuition about discriminative regions for digit recognition.

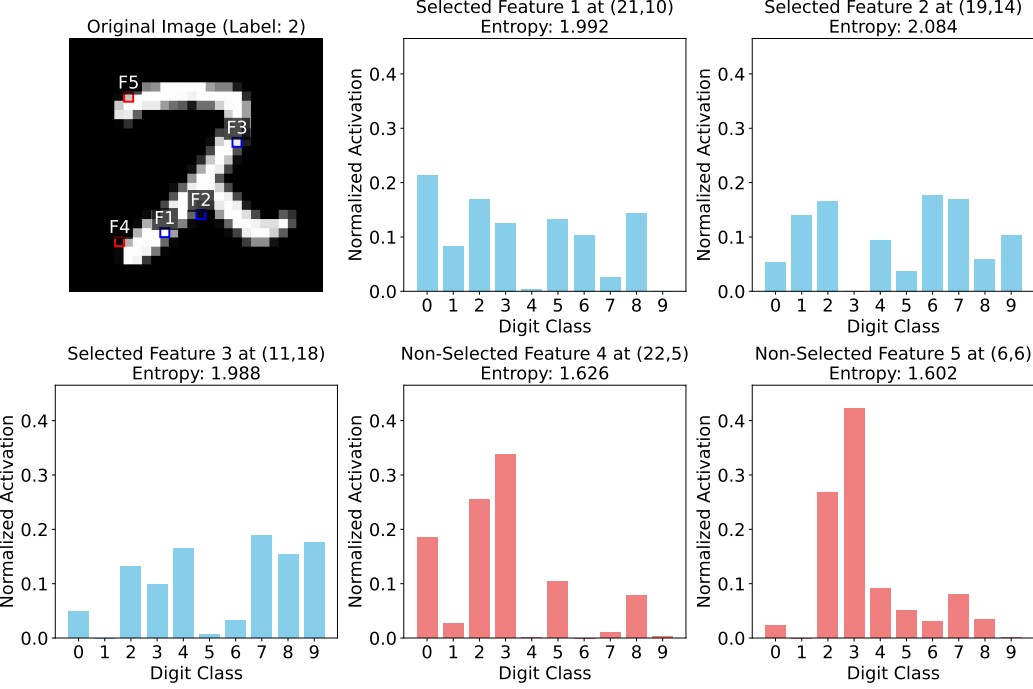

Figure 8: Analysis of sample features (top-left) from MNIST dataset shows that entropy of selected features (F1-F3) is much higher than their non-selected counterparts (F4, F5). It confirms that AutoNFS selects the most discriminative features.

