# OpenReview forum: "AutoNFS: Automatic Neural Feature Selection"
_ICLR.cc/2026/Conference — ICLR 2026 Conference Withdrawn Submission_

### Official Review · Reviewer_B7tn · 2025-10-30

**Soundness:** 2
**Presentation:** 2
**Contribution:** 2
**Rating:** 2
**Confidence:** 4

**Summary:**

The paper proposes an end-to-end learnable feature selection method for high-dimensional tabular data. Specifically, the proposed AutoNFS combines a “masking” network that uses a Gumbel-Sigmoid trick and a “target” network for learning and validating a binary mask for feature selection. The authors show several sets of experiments on standard benchmarks for feature selection (that incorporate different feature corruptions) as well as several high-dimensional metagenomic datasets. The reported results indicate that AutoNFS can correctly identify important features.

**Strengths:**

- The motivation is clear and important: having a flexible and learnable feature importance has many practical benefits, especially when applied to high-dimensional tabular learning
- Clever use of the Gumbel-Sigmoid trick for efficient learning of feature selection masks
- Clearly explained method

**Weaknesses:**

- W1: IMO, the biggest weakness of the work is the novelty. Specifically, the main contribution of this work is the use of Gumbel-Sigmoid, a binary relaxation Gumbel-Softmax, which has already been used and extensively studied for differentiable feature selection, as part of different methods such [1-3].  Even Gumbel-Softmax has been used in a similar setting [4]. The authors state these similarities, but this is never further evaluated.
- W2: The experimental analysis largely builds on benchmark data, comparing AutoNFS to several established methods. However, there are many methods, such as [1-4] as well as [5,6], etc (some of which even discussed by the authors) that have been omitted from the analysis.
- W3: The presented analysis feels incomplete and not properly documented. Feature selection/importance (beyond removing noisy/redundant features) is largely coupled to the learning mechanism of the downstream predictor. In other words, what neural network might consider useful may be different from how a tree-based model will utilize the features [7]. With this in mind, AutoNFS is tuned to the “target” network and its capabilities. This hasn’t been really discussed. The first set of experiments (Appendix D ) seems to use this feature selection with other methods, but the “native” performance of the methods is not reported. In the second, metagenomic, set of experiments, this seems to be evaluated (I assume that’s what “RF full data” means), but the tuning and parameters of RF are not provided.
- Other:
    - In several figures, AutoNFS appears to be omitted (Figure 2, Figure 3a, and Figure 4), but the authors discuss GFSNetwork? Also, Figure 3 discusses another set of FS methods that are not in the other experiments.
    - It seems the anonymous repo is broken, and I can't access it.
    - There are some references that are repeated (eg. Tibshirani et al., Yamada et al etc)

[1] Singh et al. “ Fsnet: Feature selection network on high-dimensional biological data” 2020

[2] Abid et. al “Concrete autoencoders for differentiable feature selection and reconstruction” 2020  ( **note:** The authors should correct this reference in their paper.)

[3] Borisov et. al  Cancelout: A layer for feature selection in deep neural networks.2019

[4] Mansouri et al.”A Deep Explainable Model for Fault Prediction Using IoT Sensors” 2022

[5] Chen et al “ Learning to Explain: An Information-Theoretic Perspective on Model Interpretation” 2018

[6] Yoon et al “ INVASE: Instance-wise Variable Selection using Neural Networks” 2023

[7] Grinsztajn et al. "Why do tree-based models still outperform deep learning on tabular data?” 2022

[8] Jiang et al. “ProtoGate: Prototype-based Neural Networks with Global-to-local Feature Selection for Tabular Biomedical Data” 2024

**Questions:**

- See weaknesses
- AutoNFS is a global feature selection method. There are several methods, like [5,6], that consider instance-wise (local) selection or combine both [8] with improved performance. The authors should justify their motivation. Extending AutoNFS to accommodate such scenarios may further increase its utility.
- How sensitive is the feature selection to the hyperparameters of the target network? Does it change with the architecture? What about wrt convergence?

---

### Official Review · Reviewer_rdyY · 2025-10-31

**Soundness:** 1
**Presentation:** 1
**Contribution:** 2
**Rating:** 0
**Confidence:** 5

**Summary:**

The paper proposes AutoNFS, a neural feature-selection framework that couples (i) a masking network that produces a feature mask via temperature-controlled Gumbel–Sigmoid sampling and (ii) a task network that evaluates the masked inputs end-to-end. A sparsity penalty on the (soft) mask is used to let the model “discover” how many features to retain, so the number of selected features is not a user hyperparameter. The authors claim two main advantages: (1) nearly constant computational overhead w.r.t. input dimensionality, and (2) automatic determination of a minimal feature subset. Experiments are run on the Cherepanova et al. (2023) feature-selection benchmark (11 OpenML-like datasets, three corruption scenarios) and on 24 metagenomic datasets, where AutoNFS is reported to outperform a set of classical FS baselines while selecting far fewer features. Code is said to be shared via an anonymous 4open link, but the link does not work on the reviewer side.

*  Valeriia Cherepanova, Roman Levin, Gowthami Somepalli, Jonas Geiping, C Bayan Bruss, Andrew G Wilson, Tom Goldstein, and Micah Goldblum. A performance-driven benchmark for feature selection in tabular deep learning. Advances in Neural Information Processing Systems, 36:41956–41979, 2023.

**Strengths:**

*  Clear core idea: using a single global Gumbel-Sigmoid-based mask, annealed over training, plus a task network, is a clean and reproducible recipe. The training loop in Algorithm 1 is easy to reimplement.

*  End-to-end differentiability: the paper correctly exploits Gumbel/Concrete tricks to make FS learnable with SGD, which aligns with a large body of recent neural FS / L0 / STG work. This makes the method potentially pluggable into existing tabular pipelines.

*  Automatic feature-count discovery: removing the need to fix $k$ is a real pain point for many FS users, and the paper gives a plausible mechanism ($task loss + \lambda·selection loss$) to achieve it.

*  Complexity experiment: the scaling plot claiming $~O(D^0.08)$ vs clearly worse baselines is interesting, and if reproducible, could be a selling point for very wide inputs.

* Application to metagenomic data: showing the same machinery on biomedical tabular data is nice, and it’s good to see that a large fraction of features can be dropped without collapsing accuracy.

**Weaknesses:**

* Code / reproducibility issue: the anonymous GitHub/4open link does not load (“the file does not exist”), so at the moment the paper is not reproducible.

* Outdated and incomplete related work: the introduction mostly rehashes the classic filter/wrapper/embedded taxonomy and older sparsity work, but the paper does not engage with several neural FS / selector-network / explanation-as-selection baselines that are now standard: L2X [1], INVASE [2], REAL-X [3], ProtoGate [4], LSPIN/LLSPIN [5], more recent differentiable subset-selection and masked-tabular models, and even strong modern tabular DL models that implicitly do selection/attend-over-columns. This makes the “we are new” claim weaker.

* Questionable “high-dimensional” positioning: in Table 1, almost all datasets have $≤136$ features, and one of them has 8 features, yet the paper frames AutoNFS as “for high-dimensional tabular data.” Later, the metagenomics part calls 308-718 features “high-dimensional.” For current tabular FS literature, and certainly for ICLR, this is a weak regime; the paper should show 1K-10K+ feature tasks or at least acknowledge the gap. If you truly look for High-Dimensional datasets, you can check this site for HDLSS groups: https://jundongl.github.io/scikit-feature/datasets

* Synthetic inflation of dimensionality: the paper says that when they did not have enough features, they added artificial/corrupted/second-order features and then demonstrated that AutoNFS can remove them. That only shows the method can prune its own injected noise, not that it works on naturally high-D data. The math/justification for “first create features, then prove we can select from them” is not convincing.

* Weak baselines for a tabular DL paper: all comparisons are to 10 FS methods with an MLP as downstream model. For ICLR 2026, a tabular method should be compared to strong tabular baselines (FT-Transformer [6], SAINT [7], TabTransformer [8], TabPFN [9] / TabPFN v2 [10], Trompt [11], NDTF [12], DANets [13], TabSeq [14], MambaTab [15], T2G-Former [16], TabNet [17], TabR [18], TabM [19], TabICL [20], NODE [21], TANGOS [22], AutoInt [23], DCN [24], modernNCA [25], TabulaRNN [26], DeepFM [28] etc.) and to neural selector baselines (STG [27], INVASE [2], L2X [1], REAL-X [3], ProtoGate [4], LSPIN [5]). Right now the experimental section does not prove superiority in the regime the authors are claiming.

* Unsubstantiated hyperparameter choices: $\lambda$ is fixed to 1 “because it works,” and the temperature starts at 2.0 with exponential decay, again without sensitivity analysis or theoretical guidance. For a method whose key claim is “we automatically discover the number of features,” the balance parameter should be examined much more carefully.

* Architecture figure and formatting issues: Fig. 1 does not clearly show input/output flow, nor dimensions, nor how the global mask is shared; later the paper references a figure whose caption is misplaced (line ~431), which signals poor polish.

* Experimental reporting is incomplete: no info on hardware, training time per dataset, number of runs/seeds, or statistical significance tests. All results look like single-run numbers; for FS papers, variance can be big.

* FT-Transformer citation is not connected: the paper cites FT-Transformer [6] but does not explain the relationship to feature selection/column gating in that model.

### References

* [1]. Chen, J., Song, L., Wainwright, M., & Jordan, M. (2018, July). Learning to Explain: An Information-Theoretic Perspective on Model Interpretation. In International Conference on Machine Learning (pp. 883-892). PMLR.
* [2]. Yoon, J., Jordon, J., & Van der Schaar, M. (2018, September). INVASE: Instance-wise variable selection using neural networks. In International conference on learning representations.
* [3]. Jethani, N., Sudarshan, M., Aphinyanaphongs, Y., & Ranganath, R. (2021, March). Have We Learned to Explain?: How Interpretability Methods Can Learn to Encode Predictions in Their Interpretations. In International Conference on Artificial Intelligence and Statistics (pp. 1459-1467). PMLR.
* [4]. Jiang, X., Margeloiu, A., Simidjievski, N., & Jamnik, M. (2024, July). ProtoGate: Prototype-based Neural Networks with Global-to-local Feature Selection for Tabular Biomedical Data. In International Conference on Machine Learning (pp. 21844-21878). PMLR.
* [5]. Yang, J., Lindenbaum, O., & Kluger, Y. (2022, June). Locally Sparse Neural Networks for Tabular Biomedical Data. In International Conference on Machine Learning (pp. 25123-25153). PMLR.
* [6]. Gorishniy, Y., Rubachev, I., Khrulkov, V., & Babenko, A. (2021). Revisiting Deep Learning Models for Tabular Data. Advances in Neural Information Processing Systems, 34, 18932-18943.
* [7]. Somepalli, G., Schwarzschild, A., Goldblum, M., Bruss, C. B., & Goldstein, T. SAINT: Improved Neural Networks for Tabular Data via Row Attention and Contrastive Pre-Training. In NeurIPS 2022 First Table Representation Workshop.
* [8]. Huang, X., Khetan, A., Cvitkovic, M., & Karnin, Z. (2020). TabTransformer: Tabular Data Modeling Using Contextual Embeddings. arXiv preprint arXiv:2012.06678.
* [9]. Hollmann, N., Müller, S., Eggensperger, K., & Hutter, F. TabPFN: A Transformer That Solves Small Tabular Classification Problems in a Second. In NeurIPS 2022 First Table Representation Workshop.
* [10]. Hollmann, N., Müller, S., Purucker, L., Krishnakumar, A., Körfer, M., Hoo, S. B., ... & Hutter, F. (2025). Accurate Predictions on Small Data with A Tabular Foundation Model. Nature, 637(8045), 319-326.
* [11]. Chen, K. Y., Chiang, P. H., Chou, H. R., Chen, T. W., & Chang, T. H. (2023, July). Trompt: Towards a Better Deep Neural Network for Tabular Data. In International Conference on Machine Learning (pp. 4392-4434). PMLR.
* [12]. Kontschieder, P., Fiterau, M., Criminisi, A., & Bulo, S. R. (2015). Deep Neural Decision Forests. In Proceedings of the IEEE International Conference on Computer Vision (pp. 1467-1475).
* [13]. Chen, J., Liao, K., Wan, Y., Chen, D. Z., & Wu, J. (2022, June). DANets: Deep Abstract Networks for Tabular Data Classification and Regression. In Proceedings of the AAAI Conference on Artificial Intelligence (Vol. 36, No. 4, pp. 3930-3938).
* [14]. Habib, A. Z. S. B., Wang, K., Hartley, M. A., Doretto, G., & A. Adjeroh, D. (2024, November). TabSeq: A Framework for Deep Learning on Tabular Data via Sequential Ordering. In International Conference on Pattern Recognition (pp. 418-434). Cham: Springer Nature Switzerland.
* [15]. Ahamed, M. A., & Cheng, Q. (2024, August). MambaTab: A Plug-and-Play Model for Learning Tabular Data. In 2024 IEEE 7th International Conference on Multimedia Information Processing and Retrieval (MIPR) (pp. 369-375). IEEE.
* [16]. Yan, J., Chen, J., Wu, Y., Chen, D. Z., & Wu, J. (2023, June). T2G-Former: Organizing Tabular Features into Relation Graphs Promotes Heterogeneous Feature Interaction. In Proceedings of the AAAI Conference on Artificial Intelligence (Vol. 37, No. 9, pp. 10720-10728).
* [17]. Arik, S. Ö., & Pfister, T. (2021, May). TabNet: Attentive Interpretable Tabular Learning. In Proceedings of the AAAI Conference on Artificial Intelligence (Vol. 35, No. 8, pp. 6679-6687).
* [18]. Gorishniy, Y., Rubachev, I., Kartashev, N., Shlenskii, D., Kotelnikov, A., & Babenko, A. TabR: Tabular Deep Learning Meets Nearest Neighbors. In The Twelfth International Conference on Learning Representations.
* [19]. Gorishniy, Y., Kotelnikov, A., & Babenko, A. TabM: Advancing Tabular Deep Learning with Parameter-efficient Ensembling. In The Thirteenth International Conference on Learning Representations.
* [20]. Qu, J., Holzmüller, D., Varoquaux, G., & Le Morvan, M. (2025, July). TabICL: A Tabular Foundation Model for In-Context Learning on Large Data. In ICML 2025-Forty-Second International Conference on Machine Learning.
* [21]. Popov, S., Morozov, S., & Babenko, A. Neural Oblivious Decision Ensembles for Deep Learning on Tabular Data. In International Conference on Learning Representations.
* [22]. Jeffares, A., Liu, T., Crabbé, J., Imrie, F., & van der Schaar, M. TANGOS: Regularizing Tabular Neural Networks through Gradient Orthogonalization and Specialization. In The Eleventh International Conference on Learning Representations.
* [23]. Song, W., Shi, C., Xiao, Z., Duan, Z., Xu, Y., Zhang, M., & Tang, J. (2019, November). AutoInt: Automatic Feature Interaction Learning via Self-Attentive Neural Networks. In Proceedings of the 28th ACM International Conference on Information and Knowledge Management (pp. 1161-1170).
* [24]. Wang, R., Fu, B., Fu, G., & Wang, M. (2017). Deep & Cross Network for Ad Click Predictions. In Proceedings of the ADKDD'17 (pp. 1-7).
* [25]. Ye, H. J., Yin, H. H., & Zhan, D. C. (2024). Modern Neighborhood Components Analysis: A Deep Tabular Baseline Two Decades Later. arXiv e-prints, arXiv-2407.
* [26]. Thielmann, A. F., & Samiee, S. (2024, December). On the Efficiency of NLP-Inspired Methods for Tabular Deep Learning. In NeurIPS Efficient Natural Language and Speech Processing Workshop (pp. 532-539). PMLR.
* [27]. Yamada, Y., Lindenbaum, O., Negahban, S., & Kluger, Y. (2020, November). Feature Selection Using Stochastic Gates. In International Conference on Machine Learning (pp. 10648-10659). PMLR.
* [28]. Guo, H., Tang, R., Ye, Y., Li, Z., & He, X. (2017, August). DeepFM: A Factorization-Machine Based Neural Network for CTR Prediction. In Proceedings of the 26th International Joint Conference on Artificial Intelligence (pp. 1725-1731).

**Questions:**

* Code / repo availability: The anonymized GitHub/4open link currently shows “the file does not exist” on my side. Can you confirm that (i) the repo actually contains all training/evaluation scripts, (ii) the corruption/synthetic-feature generation scripts, and (iii) the exact configs used for Tables 1–2? If it was a packaging issue, please clarify how to run it end-to-end.

* Claimed target regime (high-dimensional): You position AutoNFS as a method for high-dimensional tabular data, but the main benchmark has at most 136 features, one dataset has 8 features, and the metagenomics part is 308-718 features. Why do you consider 300-700 features “high-dimensional”? Do you have results on $≥1,000-10,000$ feature microarray/omics/text-like tabular data (for example, those in the scikit-feature HDLSS collection mentioned in weakness)? If not, please justify the claim or re-scope it.

* Synthetic / generated features: Lines 324-329 mention that you generated additional (artificial) features when the original data did not have enough, and then you ran feature selection. Please explain precisely: (i) how the extra features were generated (operations, distributions, correlations)? (ii) whether they were added to all methods or only to AutoNFS? (iii) and why selecting from self-generated noise is a meaningful evaluation of feature selection?

* Baseline coverage: Right now, baselines are mostly “FS algorithm + MLP.” Why didn’t you compare to neural selector / explainer style models that are closest to your setup (STG, INVASE, L2X, REAL-X, ProtoGate, LSPIN/LLSPIN)? Are there technical reasons AutoNFS cannot be compared to them? Please provide at least a discussion and, ideally, numbers. [References mentioned in weakness]

* Tabular DL baselines: Since you frame this as a tabular deep learning model, why are there no comparisons to strong tabular DL methods (FT-Transformer, SAINT, TabTransformer, TabR, TabM, NODE, TANGOS, TabNet, TabPFN / TabPFN v2, Trompt, MambaTab, TabSeq, T2G-Former, etc.)? Even if some of them do not do explicit FS, we need to know whether your masked MLP is actually competitive. [References mentioned in weakness]

* FT-Transformer citation (Line 95): You cite FT-Transformer but the text does not make clear what aspect you are borrowing (attention as implicit selection? column gating?). Please clarify the intended connection, or remove the citation if it is not directly relevant.

* Hyperparameters of the selector: $\lambda$ is fixed to 1 and the temperature starts at 2.0. (i) Is $\lambda = 1$ empirically optimal across all datasets, or just a convenient choice? (ii) How sensitive is AutoNFS to $\lambda$ and to the temperature schedule? (iii) Can you show a plot of selected-feature count vs. $\lambda$, and accuracy vs. $\lambda$?

* “Automatic” number of features: You state that existing methods treat the number of selected features as a user hyperparameter, but many recent works learn it. Can you clarify what is truly automatic in AutoNFS compared to $L0$-regularized or Concrete/Hard-Concrete selectors? What is the exact novelty over those?

* Reliability of results: The current tables look like single-run numbers. How many seeds did you run for each dataset? What are mean $\pm$ std and the average ranks across methods? Can you report a Friedman/Nemenyi or Wilcoxon-Holm test to show significance?

* Computational complexity section: You claim near-constant scaling $(O(D^0.08))$ in Fig. 4. (i) What is the exact experimental setup for this figure (hardware, batch size, epochs)? (ii) Does this scaling hold for the baselines when implemented in the same framework? (iii) Is the cost of Gumbel sampling and temperature annealing included?

* Choice of datasets: Why was the CH dataset with only 8 features kept in a paper about feature selection? What would AutoNFS do on datasets where the optimal answer is “select everything”? Please justify.

* Missing / misplaced figure and formatting: Around Line 431 there is a caption but no figure. Was something omitted during anonymization or PDF generation? Please provide the missing figure and indicate where it belongs.

* Implementation details: Please specify optimizer(s), learning rate(s), batch size, number of epochs, early stopping, and GPU/CPU used. Without this, it’s hard to understand whether some baselines underperformed due to training budget.

* Generalization to other predictors: Right now, AutoNFS is shown mostly with an MLP-style predictor. Can the selector be plugged in front of a transformer-style tabular model, and if so, have you tried it? If not, what are the constraints?

* Scope of the claim: If you cannot provide truly high-dimensional ($1K–10K+$) experiments and stronger baselines, would you be willing to reframe AutoNFS as “a simple, differentiable FS layer for low-to-moderate dimensional tabular tasks”? That would make the paper’s claims match its evidence.

---

### Official Review · Reviewer_wtkb · 2025-10-31

**Soundness:** 2
**Presentation:** 3
**Contribution:** 1
**Rating:** 2
**Confidence:** 5

**Summary:**

The paper considers end-to-end differentiable feature selection using relaxed sampling. This is a promising approach to feature selection, allowing gradient-based optimization of feature selection in neural networks. However, it is not a novel approach. Similar methods have been explored in a number of previous papers [1-4]. More broadly, differentiable feature selection (without relaxed sampling) has also been considered in earlier papers [5-8]. With previous work taken into account, the paper does not present a significant novelty in its method and such claims should be toned down. In my view, the contribution is instead to benchmark this approach to feature selection.

Most of the benchmarks in this paper are repeated from Cherepanova et al [9], comparing the same models (with the addition of this paper's proposed model) across the same datasets (Table 1) and new datasets (Table 2). Thus, most of the benchmark results were already known.
As mentioned, there are a number of previous methods [1-8] that have yet to be compared in a larger benchmark. By including these, the paper could be extended to serve as a valuable benchmark of a promising approach to feature selection. Such a benchmark paper is valuable in its own right, and does not necessarily need to propose a new model.

In conclusion, the paper adresses an interesting research question, but neither its proposed method nor benchmark comparison is novel. I encourage the authors to extend the comparison to focus on feature selection using differentiable sampling. However, this requires significant changes to the paper, so my recommendation is to reject the paper.

**Strengths:**

1. **Relevant problem**. The problem, end-to-end differentiable feature selection, is a promising approach to an important problem in machine learning. To the best of my knowledge, there are no dedicated benchmark papers for these particular models.
2. **Datasets**. The experiments (following Cherepanova et al [9]) consider three different sets of corrupted features across multiple datasets.

**Weaknesses:**

1. **Lack of novely**. The proposed model is very similar to previous works, and there is neither a theoretical argument nor experimental evidence that this model should be prefered. I will list the similarities: Gumbel-softmax/sigmoid relaxation for feature selection [1, 2], the weighted L0 loss [3], seed embedding and network parameterization of logits [4]. Likewise, annealing the temperature was done in multiple previous works. I believe the paper currently overstates the novelty of its proposed method and should, for example, acknowledge some of the previous works in the section on line 138.
2. **Lack of similar baselines**. The previous benchmark paper of Cherepanova et al. [9] proposed a benchmark for tabular feature selection (in fact, this papers experimental setup follows that paper, as stated on line 322). The novelty of this paper is to consider end-to-end differentiable feature selection using differentiable sampling. This paper compares *the same* models as Cherepanova et al. [9], except adding the proposed GFSNetwork to the comparison.
3. **The parameter $\lambda$**. The parameter $\lambda$ that weighs the mask loss is set to the constant 1. Clearly, the impact of $\lambda$ depends on the relative scales of the task and mask losses, so this choice is not advisable in all cases. This is discussed in Appendix F, but parts of the main paper may give the wrong impression of this parameter. Furthermore, a parameter like this is present in most embedded feature selection models, including the standard lasso, so the claims of automatic feature selection are not novel.
4. **No Pareto comparison**. Feature selection can be understood as a type of model selection, where we are balancing the task and mask losses. Naturally, this trade-off gives rise to a Pareto frontier. I believe *automatic* feature selection, as discussed in the paper's abstract and introduction, should return a solution at the "elbow" of the Pareto frontier. In fact, LassoNet [5] addressed this problem. It would be natural to compare models in a "loss vs. #features" plot to see which ones lie closer to the Pareto frontier (like Figure 6 but comparing different methods).

**Questions:**

1. **Top-k**. I believe that a central question in differentiable feature selection is whether to use top-k sampling or not. Top-k sampling with learned k is possible [7], which allows for a weighted objective using a parameter $\lambda$. Compared to sampling Bernoulli and penalizing the outcome, top-k sampling would reduce the variance in the number of selected features per forward pass. Do you think a comparison with some top-k approach [2,6,7,8] is warranted?
2. **Non-differentiable models**. A limitation of end-to-end differentiable feature selection, compared to filter and wrapper methods, is that the task model must be differentiable. I think this limitation deserves mention in the context of *automatic* feature selection because it limits the choice of model to only those that are differentiable. It is possible to circumvent this problem by using a score-function estimator for the gradients [8]. Do you agree with this limitation?
4. **Downstream models**. Like in Cherepanova et al. [9], the feature selection methods are compared by training downstream models on the selected features (line 330). This is good because it makes sure to compare feature sets and not models. However, in the case off feature selection methods other than filter methods, there might be a risk of unfair comparisons. For example, consider using the same wrapper method to select features by first wrapping an MLP and then XGBoost, and then evaluating the feature set by training MLPs. The first case selected features *given* the class of model used for evaluation, while the second case used a different class of models. Do you agree that there is a risk of favoring MLP models?
5. **Time complexity**. Why are the time complexities estimated empirically instead of derived analytically? Of course, it gives a practical comparison, but it is implementation-dependent and seems more complicated than theoretical time complexities.

**Minor comments**
- Citations are not wrapped in parentheses. Consider using, e.g. \citep.
- Many references appear twice, which should be easy to fix.
- Line 453 says “Mutual Informatio”, missing “n”.
- On line 135, the task network is likened to a discriminator. I find this comparison confusing, as the training setup is nothing like a GAN.

**References**

[1] Balın et al. "Concrete Autoencoders: Differentiable Feature Selection and Reconstruction". ICML 2019.

[2] Huijben et al. "Deep Probabilistic Subsampling for Task-Adaptive Compressed Sensing". ICLR 2019.

[3] Yamada et al. "Feature Selection using Stochastic Gates". ICML 2020.

[4] Nilsson et al. "Indirectly Parameterized Concrete Autoencoders". ICML 2024.

[5] Lemhadri et al. "LassoNet: Neural Networks with Feature Sparsity". AISTATS 2021.

[6] Ahmed et al. "SIMPLE: A Gradient Estimator for k-Subset Sampling". ICLR 2023.

[7] Pervez et al. "Scalable Subset Sampling with Neural Conditional Poisson Networks". ICLR 2023.

[8] Wijk et al. "SFESS: Score Function Estimators for k-Subset Sampling". ICLR 2025.

[9] Cherepanova et al. "A Performance-Driven Benchmark for Feature Selection in Tabular Deep Learning". NeurIPS 2023.

---

> ### Author Response · Authors · 2025-12-02
>
> We appreciate the reviewer’s thoughtful questions. We will adress the most important ones below.
>
>
> > Lack of novely.
>
> ​​We agree that AutoNFS builds on prior ideas such as relaxed sampling, L0-style sparsity, and temperature annealing, and we will acknowledge these works more explicitly. The contribution of AutoNFS is not a new relaxation, but a global, dataset-level feature selector with mask parameterization independent of D, unlike STG, Concrete AE, or INVASE, which use per-feature or instance-wise gating. This enables the near-constant scaling and stable emergent cardinality shown in the experiments.
>
> > Lack of similar baselines.
>
> We fully agree.
>  Our initial goal was to evaluate AutoNFS within the established 2023 benchmark, ensuring comparability of results across the same dataset suite and corruption protocols. However, we agree that including differentiable sampling methods is important for a complete picture. In the revision, we are adding baselines.
>
> > The parameter lambda
>
> We clarify the following:
> - λ is not claimed to be theoretically universal. We performed a sensitivity analysis in Fig. 6, showing that λ=1 yields a stable operating point across all datasets in the benchmark.
> - AutoNFS does not require specifying k. Even with a sparsity penalty, the number of active features is not a hyperparameter and emerges from optimization - unlike LASSO, LassoNet, and many embedded FS methods that require selecting a budget or tuning λ extensively.
>
> > No Pareto comparison.
>
> - We agree that Pareto analysis is valuable.  In practice, the full Pareto frontier is very expensive to compute even for medium-sized datasets - for wrapper methods it requires training N models for N possible cardinalities. For AutoNFS, it requires sweeping λ over dense grids.
>
> > Top-k.
>
> We agree that top-k–based subset sampling is an important line of work and that methods such as SIMPLE, CPoisson networks, and SFESS address a complementary problem. Our focus in AutoNFS is deliberately different: we aim for a global selector with emergent, not fixed, cardinality, where the number of selected features is not constrained to k but arises from the trade-off with the task loss.
>
> Top-k methods enforce a hard budget per forward pass, while AutoNFS allows the model to decide how many features it needs. Because of this difference in objective (fixed-k vs. emergent-k) and the computational cost of integrating multiple k-subset samplers into the full Cherepanova benchmark, we treated a systematic comparison to top-k approaches as out of scope for this work. We will clarify this in the paper and highlight top-k sampling as a natural direction for future extensions of AutoNFS.
>
> > Non-differentiable models.
>
> You are correct that end-to-end FS requires a differentiable task model during training.
> However:
> - AutoNFS is designed as a selector, not as a predictor.
> - Once the mask is learned, any downstream model - including non-differentiable ones - can be trained on the selected subset.
>
> This is demonstrated in Table 2, where feature subsets learned using MLP improve performance of Random Forests, which are fully non-differentiable.
> Thus, the method’s limitation applies only to the training phase, not the downstream evaluation.
>
> > Downstream models. Like in Cherepanova et al. [9], the feature selection methods are compared by training downstream models on the selected features (line 330). This is good because it makes sure to compare feature sets and not models. However, in the case off feature selection methods other than filter methods, there might be a risk of unfair comparisons. For example, consider using the same wrapper method to select features by first wrapping an MLP and then XGBoost, and then evaluating the feature set by training MLPs. The first case selected features given the class of model used for evaluation, while the second case used a different class of models. Do you agree that there is a risk of favoring MLP models?
>
> We agree this risk exists in principle.
> However, in Table 2 we explicitly evaluate the selected subsets on both MLP and RF, and in both cases AutoNFS improves performance - indicating that the selected features generalize across model families.
>
> > Time complexity.
> Why are the time complexities estimated empirically instead of derived analytically? Of course, it gives a practical comparison, but it is implementation-dependent and seems more complicated than theoretical time complexities.
>
> You are correct that theoretical complexities could be derived.
>  However, for GPU-accelerated tabular models, analytical FLOP counts poorly correlate with wall-clock costs, because actual GPU execution depends heavily on kernel fusion and parallelism.
> Empirical GPU time is therefore the most informative and standard measure (similar to TabPFN, SAINT, FT-Transformer evaluations).

---

### Official Review · Reviewer_H8PF · 2025-10-31

**Soundness:** 2
**Presentation:** 3
**Contribution:** 2
**Rating:** 2
**Confidence:** 4

**Summary:**

The paper introduces AutoNFS, a fully differentiable, end-to-end neural network framework for automatic feature selection (FS). It integrates a masking network using Gumbel–Sigmoid sampling with a task network that evaluates selected features’ relevance. AutoNFS automatically determines the minimal sufficient subset of features for a downstream task, eliminating the need for a user-specified feature count.

**Strengths:**

1, High computational efficiency with nearly constant time overhead.
2, End-to-end differentiable training.
3, Applicable to high-dimensional data and diverse downstream models.

**Weaknesses:**

1, The core claim that AutoNFS can automatically determine the optimal number of features, lacks any theoretical justification. The penalty term is simply an L1-style sparsity regularization. There is no theoretical guarantee that this formulation leads to a globally optimal or stable feature subset.
2, The method combines Gumbel–Sigmoid relaxation with a sparsity penalty, a concept already established in STG (Yamada et al., ICML 2020), INVASE (Yoon et al., ICLR 2018).  AutoNFS does not introduce a fundamentally new formulation, optimization objective, or architectural insight beyond these works.
3, Most baselines are outdated (e.g., Lasso, RF, ANOVA F-value). The absence of recent differentiable or transformer-based FS methods biases the results.
4, There is no analysis of whether the selected features correspond to meaningful attributes.
5, The sparsity weight λ is always set to 1 without sensitivity analysis.

**Questions:**

1, How robust is the model’s performance to variations in λ across datasets?
2, Why was an exponential decay with α = 0.997 chosen?
3, How does AutoNFS compare with Transformer-based or contrastive FS approaches ?
4, How consistent are the selected features across different random initializations or data splits?

---

> ### Author Response · Authors · 2025-12-02
>
> We thank the reviewer for their insightful comments. Below, we will address the most critical ones.
>
> ---
>
> >The core claim that AutoNFS can automatically determine the optimal number of features, lacks any theoretical justification. The penalty term is simply an L1-style sparsity regularization. There is no theoretical guarantee that this formulation leads to a globally optimal or stable feature subset.
>
> Thank you for raising this. Our claim is empirical, not theoretical:
>  for a fixed value of λ, AutoNFS consistently converges to a stable number of active features. This behavior is documented in Fig. 6, where:
> λ=1 yields a clear optimum for both accuracy and feature-count stability across datasets,
> the number of selected features emerges naturally during optimization without requiring a manually specified budget k.
> We do not claim a formal global optimality guarantee (none of the related differentiable FS methods - STG, Hard Concrete, INVASE - provide one either), and we will clarify this distinction explicitly in the revision. The key point is that the combination of global masking + annealed Gumbel-Sigmoid + sparsity consistently converges to a dataset-level cardinality without user tuning.
>
> > The method combines Gumbel–Sigmoid relaxation with a sparsity penalty, a concept already established in STG (Yamada et al., ICML 2020), INVASE (Yoon et al., ICLR 2018). AutoNFS does not introduce a fundamentally new formulation, optimization objective, or architectural insight beyond these works.
>
> We agree that combining a continuous relaxation with a sparsity penalty is not new—STG, Hard-Concrete/L0, and INVASE all build on this general idea. AutoNFS is not intended to introduce a new relaxation, but rather to provide a different formulation tailored to global (dataset-level) feature selection, which these methods do not directly address.
> Specifically:
> - Global mask instead of per-instance or per-feature gating. STG and Hard-Concrete maintain a gate per feature, and INVASE produces instance-wise masks. AutoNFS learns one global mask that identifies a stable feature subset—this is the regime evaluated in the Cherepanova benchmark and required in metagenomic biomarker settings.
> - Complexity that does not scale with the number of features. Prior methods incur O(D) gating costs. AutoNFS uses a compact mask network whose cost is independent of D, enabling the near-constant scaling we report.
> - Stable emergent cardinality under annealed Gumbel-Sigmoid. AutoNFS does not require specifying k or tuning per-feature gates; the number of selected features emerges naturally (Fig. 6)
>
>
> Thus, the novelty of AutoNFS lies not in inventing a new relaxation, but in providing a scalable, global, cardinality-emergent feature selector - which, to our knowledge, is not addressed by STG, Hard-Concrete, or INVASE.
>
> > Most baselines are outdated (e.g., Lasso, RF, ANOVA F-value). The absence of recent differentiable or transformer-based FS methods biases the results.
>
> We intentionally follow the Cherepanova et al. (2023) benchmark to ensure directly comparable results. This benchmark includes classical FS baselines, and an MLP-based task network for all methods.
> That being said, we fully agree that differentiable and transformer-based FS baselines should be included to strengthen the paper.
>
> > There is no analysis of whether the selected features correspond to meaningful attributes.
>
> Figure 3a directly addresses this:
> AutoNFS selects biologically important features in the metagenomic datasets, and these selected subsets improve downstream performance across multiple models (MLP, RF).
>
> > The sparsity weight λ is always set to 1 without sensitivity analysis.
>
> A λ-sensitivity analysis is included in Fig. 6.
>
> Questions:
> > How robust is the model’s performance to variations in λ across datasets?
>
> The robustness analysis is provided in Fig. 6.
>  λ=1 consistently provides the best trade-off across datasets, while lower/higher values either under- or over-prune.
>
> > Why was an exponential decay with α = 0.997 chosen?
>
> We used exponential decay because:
> - it is the standard approach in temperature-annealed Gumbel sampling,
> - α=0.997 provided stable convergence across all datasets in preliminary experiments.
>
>
> > How does AutoNFS compare with Transformer-based or contrastive FS approaches ?
>
> We agree that such baselines are valuable.
> We will be adding comparisons with recent transformer-based tabular models and contrastive tabular methods where feasible (e.g., FT-Transformer, SAINT, RealMLP-based selectors, and strong modern MLPs).
> If direct FS-enabled versions are unavailable, we will still test whether these models benefit from AutoNFS-selected subsets.
>
> > How consistent are the selected features across different random initializations or data splits?
>
> We are running multi-seed evaluations and will report:
> - feature-set overlap across seeds,
> - Jaccard similarity,
> - variance in selected cardinality.

---

### Official Review · Reviewer_xaMc · 2025-11-01

**Soundness:** 2
**Presentation:** 2
**Contribution:** 2
**Rating:** 4
**Confidence:** 5

**Summary:**

This paper proposes an end-to-end feature selection method using Gumbel–Sigmoid masks with a sparsity penalty to automatically determine both which and how many features to keep. The paper claims near-constant computational cost with respect to input dimensionality and reports strong results on OpenML and metagenomic datasets.

**Strengths:**

The use of Gumbel–Sigmoid masks for feature selection is a good idea.

**Weaknesses:**

The authors should provide stronger evidence that the proposed method outperforms well-tuned baselines.

**Questions:**

1. For regression tasks, $R^2$ would be a more appropriate evaluation metric than MSE. MSE is sensitive to the scale of the target variable and may not be informative for comparing models.
2. The citation format does not follow the ICLR style. Please revise accordingly.
3. It is unclear why second-order interaction features are treated as incorrect. In tabular learning, many works aim to enable neural networks to capture higher-order feature interactions. Please clarify this choice.
4. In Table 2, improvements from the proposed method on algorithms such as MLP and RF are shown only for the metagenomic benchmark. It remains unclear how well the method generalizes to general tabular learning tasks. Please include results on OpenML benchmarks.
5. In Table 2, only a comparison between the full feature set and AutoNFS-selected features is provided. Please also include comparisons with other feature selection baselines already shown in Figure 2.
6. The MLP used in Table 2 appears to be a standard version without extensive hyperparameter tuning. Please consider using RealMLP [1] as a stronger baseline.
7. In Table 2, include the average rank of methods for improved clarity and consistency.

[1] Holzmüller, D., Grinsztajn, L., & Steinwart, I. (2024). Better by default: Strong pre-tuned MLPs and boosted trees on tabular data. NeurIPS 37: 26577–26658.

---

> ### Author Response · Authors · 2025-12-02
>
> > For regression tasks, R^2 would be a more appropriate evaluation metric than MSE. MSE is sensitive to the scale of the target variable and may not be informative for comparing models.
>
> The benchmark we follow (Cherepanova et al. 2023) uses MSE as the primary metric for regression tasks.
> To preserve comparability with published results, we report MSE in the main tables.
>
> > It is unclear why second-order interaction features are treated as incorrect. In tabular learning, many works aim to enable neural networks to capture higher-order feature interactions. Please clarify this choice.
>
> Second-order interaction features in our experiments come directly from the Cherepanova et al. benchmark protocol.
> In this setting:
> - these interaction features are synthetic perturbations intentionally injected to stress-test feature selectors,
> - the goal is to evaluate robustness to corrupted or misleading features,w
> We agree that higher-order interactions are important in tabular learning in general, but in the context of this benchmark they form part of the corruption component of the evaluation, not part of the true signal.
> We will clarify this distinction in the revised text and explicitly state that we are following the benchmark protocol rather than asserting that interaction features are inherently “incorrect”.
>
> > In Table 2, improvements from the proposed method on algorithms such as MLP and RF are shown only for the metagenomic benchmark. It remains unclear how well the method generalizes to general tabular learning tasks. Please include results on OpenML benchmarks.
>
> Table 2 focuses on metagenomic datasets because our goal in this section was to provide a proof-of-concept demonstrating that a single global mask learned by AutoNFS improves downstream performance, even when evaluated with a completely different predictor than the one used during training. For this reason, we intentionally restricted the evaluation to one differentiable model (MLP) for mask learning and validated the resulting feature subset using both MLP and RF.
> The purpose of this experiment was not to exhaustively benchmark all predictors on OpenML, but to show that:
> - AutoNFS can identify biologically important global features, and
> - these features remain useful when transferred to a non-differentiable model (RF).
> We agree that extending this evaluation to additional predictors and diverse OpenML datasets is valuable, but this goes beyond the scope of the current proof-of-concept. We will clarify this intention in the revision.
>
> > The MLP used in Table 2 appears to be a standard version without extensive hyperparameter tuning. Please consider using RealMLP [1] as a stronger baseline.
>
> We agree that RealMLP is a strong baseline. In this paper, however, our experiments were designed as a proof-of-concept focused on the behavior of AutoNFS rather than on optimizing the downstream predictor. For consistency with the Cherepanova benchmark, we used a standard MLP. Extending the evaluation to architectures like RealMLP is possible, but falls outside the scope of this proof-of-concept setup.

---

### Official Review · Reviewer_JRGE · 2025-11-03

**Soundness:** 2
**Presentation:** 3
**Contribution:** 2
**Rating:** 4
**Confidence:** 3

**Summary:**

The paper introduces AutoNFS, an end-to-end neural feature selection framework that couples a Gumbel-Sigmoid–based masking module with a task network. A sparsity penalty encourages discovery of a minimal, task-sufficient subset of features, and a temperature-annealed sampling schedule transitions from soft to nearly binary masks. The method claims nearly constant selection-time overhead with respect to input dimensionality and automatically determines the number of features to retain. Across OpenML benchmarks with synthetic corruptions and 24 metagenomic datasets, AutoNFS reports superior or competitive predictive performance while selecting substantially fewer features, along with empirical scaling advantages.

**Strengths:**

1. The work targets automatic determination of both which features and how many to select, addressing a common pain point in FS pipelines that typically require budget tuning and repeated retraining.
2. The Gumbel-Sigmoid masking with temperature annealing and a single sparsity term yields an end-to-end, SGD-trainable objective that is easy to implement and integrate with standard predictors.
3. The selection-time scaling study indicates a near-constant exponent (≈0.08) versus linear or superlinear trends for common baselines, supporting scalability claims for high-dimensional inputs.

**Weaknesses:**

1. Hard-Concrete L0, STG, Concrete Autoencoders, and INVASE also learn discrete/sparse selections end-to-end and can infer feature counts via regularization; the specific novelty beyond using Gumbel-Sigmoid with a global mask and annealing may be limited.
2. A single mask for all samples ignores instance-specific relevance; prior methods provide per-instance selection, which can be crucial under heterogeneity or conditional feature utility.
3. The selection-time metric and measurement protocol are not fully specified; the task network’s first layer scales with D, and comparisons may conflate FS overhead with predictor costs. Stronger wall-clock and FLOP-normalized studies and fairness controls are needed.
4. Several strong neural FS baselines most closely related to the proposed method (STG, L0/Hard-Concrete, Concrete AE, INVASE) are absent from accuracy and scaling comparisons. Moreover, comparisons often use mismatched feature budgets, favoring the proposed method’s auto-sparsity.

**Questions:**

Please refer to the weaknesses section.

---

> ### Author Response · Authors · 2025-12-02
>
> We thank the reviewer for their insights and comments. We will reply to the most important concerns below:
>
> > Hard-Concrete L0, STG, Concrete Autoencoders, and INVASE also learn discrete/sparse selections end-to-end and can infer feature counts via regularization; the specific novelty beyond using Gumbel-Sigmoid with a global mask and annealing may be limited.
>
> ​​We appreciate the reviewer highlighting these related methods. AutoNFS indeed builds on the general idea of differentiable sparsification, but it differs from Hard-Concrete L0, STG, Concrete Autoencoders, and INVASE in several key aspects relevant to classical feature selection:
>
> 1. Global vs. instance-wise selection. INVASE, L2X, and related approaches produce instance-specific masks; STG and Concrete AE use per-feature gates. AutoNFS instead learns one global, dataset-level mask, which is the setting required in tabular FS benchmarks and in metagenomic data classification.
> 2. Mask parameterization independent of dimensionality. Prior methods maintain a gate per feature and therefore scale linearly in D. AutoNFS uses a compact global masking network, whose compute and parameter count remain essentially constant as D grows - enabling the near-constant scaling observed empirically.
> 3. Annealed Gumbel–Sigmoid for stable emergent cardinality. AutoNFS does not require specifying k or performing top-k projection; the number of selected features emerges naturally from training
>
> Our goal is not to claim a fundamentally new relaxation, but a practical, scalable formulation tailored to global feature selection, which prior instance-wise or per-feature gating methods do not directly address. We will clarify this positioning in the revision.
>
> > A single mask for all samples ignores instance-specific relevance; prior methods provide per-instance selection, which can be crucial under heterogeneity or conditional feature utility.
>
> This is a valuable observation, and we fully agree that instance-wise selection is useful for tasks such as explanation or conditional utility modeling.
> However, AutoNFS is designed for a different problem:
> Our goal is to discover a single minimal subset of globally informative features.
> This is the setting used the Cherepanova benchmark, and in the metagenomic domain.
> Instance-wise selectors cannot guarantee sparsity at the dataset level and cannot directly optimize for a global feature budget.
>
> > The selection-time metric and measurement protocol are not fully specified; the task network’s first layer scales with D, and comparisons may conflate FS overhead with predictor costs. Stronger wall-clock and FLOP-normalized studies and fairness controls are needed.
>
> We thank ther reviewer for this comment. To clarify:
> All scaling comparisons were performed on the same hardware (1×A100 80GB).
> The reported times include all components, including Gumbel sampling and temperature annealing.
> Because the mask is a trainable parameter shared across all data, its forward cost is constant in D.
>
> > Several strong neural FS baselines most closely related to the proposed method (STG, L0/Hard-Concrete, Concrete AE, INVASE) are absent from accuracy and scaling comparisons. Moreover, comparisons often use mismatched feature budgets, favoring the proposed method’s auto-sparsity.
>
> We followed the Cherepanova et al. benchmark, which standardizes:
> - the corruption protocols,
> - the dataset suite,
> - the set of FS baselines (forward selection, MI, Relief, LASSO, etc.).
>
> This ensures clean comparability to the currently most widely used FS benchmark in deep learning.
> That being said, we fully agree that including neural selector baselines is valuable - especially STG, Hard-Concrete, and INVASE.
>
> Finally, AutoNFS does not rely on artificially favorable budget settings:
> The mask learns the number of active features end-to-end;
> Competing methods with fixed budgets were run with the optimized hyperparameters in the original benchmark.

---

### Note · Authors · 2026-01-21

**Comment:**

We thank the reviewers for their effort and their feedback, which we will use in further improving the paper.

**Withdrawal Confirmation:**

I have read and agree with the venue's withdrawal policy on behalf of myself and my co-authors.